# Tween 80-Based Self-Assembled Mixed Micelles Boost Valsartan Transdermal Delivery

**DOI:** 10.3390/ph17010019

**Published:** 2023-12-22

**Authors:** Alaa Eldeen B. Yassin, Salam Massadeh, Abdullah A. Alshwaimi, Raslan H. Kittaneh, Mustafa E. Omer, Dilshad Ahmad, Al Hassan Aodah, Faiyaz Shakeel, Majed Halwani, Saleh A. Alanazi, Prawez Alam

**Affiliations:** 1College of Pharmacy, King Abdullah International Medical Research Center, King Saud Bin Abdulaziz University for Health Sciences, Riyadh 11481, Saudi Arabiaanazis@mngha.med.sa (S.A.A.); 2Developmental Medicine Department, King Abdullah International Medical Research Center, King Saud Bin Abdulaziz University for Health Sciences, Riyadh 11481, Saudi Arabia; massadehsa@ngha.med.sa; 3Joint Centers of Excellence Program, KACST-BWH/Harvard Center of Excellence for Biomedicine, King Abdulaziz City for Science and Technology (KACST), Riyadh 11442, Saudi Arabia; 4AstraZeneca Saudi Arabia, Riyadh 13315, Saudi Arabia; alshwaimi021@hotmail.com; 5Department of Pharmacy, Faculty of Medicine and Health Sciences, An-Najah National University, Nablus P400, Palestine; raslan.kettaneh@najah.edu; 6Pharmacy Program, College of Health and Sport Sciences, University of Bahrain, Manama 32038, Bahrain; mmomer@uob.edu.bh; 7Advanced Diagnostic and Therapeutic Institute, Health Sector, King Abdulaziz City for Science and Technology (KACST), Riyadh 11442, Saudi Arabia; aaodah@kacst.edu.sa; 8Department of Pharmaceutics, College of Pharmacy, King Saud University, Riyadh 11451, Saudi Arabia; fsahmad@ksu.edu.sa; 9Nanomedicine Department, King Abdullah International Medical Research Center, King Saud Bin Abdulaziz University for Health Sciences, Riyadh 11481, Saudi Arabia; halawanima@ngha.med.sa; 10Pharmaceutical Care Services, King Abdulaziz Medical City, National Guard Health Affairs (NGHA), Riyadh 11426, Saudi Arabia; 11Department of Pharmacognosy, College of Pharmacy, Prince Sattam Bin Abdulaziz University, Al-Kharj 11942, Saudi Arabia

**Keywords:** valsartan, thin-film hydration, Franz diffusion cell, Tween 80

## Abstract

Valsartan (Val) is an important antihypertensive medication with poor absorption and low oral bioavailability. These constraints are due to its poor solubility and dissolution rate. The purpose of this study was to optimize a mixed micelle system for the transdermal delivery of Val in order to improve its therapeutic performance by providing prolonged uniform drug levels while minimizing drug side effects. Thin-film hydration and micro-phase separation were used to produce Val-loaded mixed micelle systems. A variety of factors, including the surfactant type and drug-to-surfactant ratio, were optimized to produce micelles with a low size and high Val entrapment efficiency (EE). The size, polydispersity index (PDI), zeta potential, and drug EE of the prepared micelles were all measured. The in vitro drug release profiles were assessed using dialysis bags, and the permeation through abdominal rat skin was assessed using a Franz diffusion cell. All formulations had high EE levels exceeding 90% and low particle charges. The micellar sizes ranged from 107.6 to 191.7 nm, with average PDI values of 0.3. The in vitro release demonstrated a uniform slow rate that lasted one week with varying extents. F7 demonstrated a significant (*p* < 0.01) transdermal efflux of 68.84 ± 3.96 µg/cm^2^/h through rat skin when compared to the control. As a result, the enhancement factor was 16.57. In summary, Val-loaded mixed micelles were successfully prepared using two simple methods with high reproducibility, and extensive transdermal delivery was demonstrated in the absence of any aggressive skin-modifying enhancers.

## 1. Introduction

Valsartan (Val) is a popular antihypertensive medication that belongs to the angiotensin II receptor antagonist class. It was listed as one of the top 200 prescribed medications in the United States [1]. Valsartan (Val) is an important antihypertensive medication with poor absorption and low oral bioavailability [2,3]. These constraints are due to its poor solubility and dissolution rate. The purpose of this study was to optimize a mixed micelle system for the transdermal delivery of Val in order to improve its therapeutic performance by providing prolonged uniform drug levels while minimizing drug side effects. This has prompted many researchers to work on improving Val’s overall therapeutic performance by modifying the drug’s pharmacokinetic properties. In the treatment of chronic diseases such as hypertension, transdermal drug delivery has been seen to demonstrate significant advantages over the oral route [4]. These include highly controlled blood levels of the drugs, comparable to intravenous infusion treatment, avoiding gastric intestinal tract-related side effects, and minimizing the drug’s systemic adverse effects, as well as the possibility of reducing the frequency of administration, which leads to higher patient compliance.

Aside from the low oral bioavailability, the physicochemical properties of Val, such as its low molecular weight (435.5 D), partition coefficient (log *p* = 4.5), and pKa value of 4.75, make it a viable candidate for transdermal drug delivery [5,6,7]. However, manipulation of the skin’s protective impermeable stratum corneum layer to allow drug efflux into the cutaneous layer is regarded as critical for the successful design of a transdermal delivery system. These substances are known as skin penetration enhancers. They can disrupt the stratum corneum’s integrity through a variety of mechanisms, including changing the configuration order and/or partial extraction of intercellular lipids, resulting in mobilization, and adjusting the drug partition coefficient to increase its diffusion through the skin [8,9,10]. Many substances have been shown to increase skin permeability to many drugs, including azones, surfactants, solvents such as alcohols and dimethyl sulfoxide (DMSO), essential oils, and fatty acids such as terpenes, linoleic and oleic acids [11,12,13,14]. Drug incorporation in nano-drug delivery systems has also been used to improve cellular uptake and absorption [15,16], the aqueous solubility of poorly soluble drugs [17,18], and drug residence time in the body [19]. Many nano-particulate systems have been used to improve drug skin permeability; liposomes, solid lipid nanoparticles, transfersomes, niosomes, nano-emulsions, and mixed micelles have been successfully used to improve the transdermal penetration of many drugs [20,21,22,23,24].

Micelles are association colloids that are formed by the aggregation of amphiphilic molecules which contain polar heads and non-polar tails. They associate in water to form spherical-shaped particles in which the non-polar tails hide on the inside [19]. Thus, hydrophobic drugs can be incorporated into the hydrophobic core of micelles. The hydrophilic surface of such micellar systems also has the advantage of being intrinsically protected against removal in systemic circulation by the phagocytic immune mechanism without the need for additional modification. As a result, they have a longer in vivo residence time [25,26,27]. They also comprise additional advantages, including their simple production by self-assembly methods, and are a highly promising option for the delivery of poorly soluble drugs, enhancing their solubility and bioavailability [28]. The size of a micelle is highly dependent on the molecular size and configuration of the amphiphile [29]. Micelles have been extensively applied to enhance the solubility and bioavailability of many drugs [29,30]. Polymeric micelles are a unique type of micelle that are formed by amphiphilic block co-polymers composed of alternating hydrophilic and hydrophobic segments [31]. They can form micelles by molecular plumbing out to hide the hydrophobic segment toward the core and keep the hydrophilic segment on the shell [32]. These are also known as unimer micelles and are characterized by higher stability compared with conventional micelles [33]. Mixed micelles are micelles formed by the association of two or more species of amphiphilic compounds. Their smaller sizes (below 60 nm) and simple method of production are important advantages over vesicular bilayer systems for the delivery of poorly soluble drugs, especially through the parenteral route [34].

Micellar systems have been used to improve the parenteral delivery and anticancer efficacy of anticancer drugs such as paclitaxel [35], doxorubicin [36,37], and camptothecin [38]. Song et al. [39] discovered that D-α-tocopheryl polyethylene glycol succinate/phospholipid mixed micelles significantly enhanced the parenteral delivery and anticancer activity of icariside II in multi-resistant breast cancer cells. Ould-Ouali et al. [40] demonstrated that incorporating a poorly water-soluble drug, risperidone, into a polymeric micelle system improved its solubility and oral delivery. Mixed micelles have also been successfully used as carriers for hydrophobic drugs, such as curcumin, to enhance their absorption rate and oral delivery [41]. Some oral drug delivery systems, such as self-nano emulsifying drug delivery systems (SNEDDS), solid SNEDDS, proliposomes, and polymer-based supersaturable self-micro emulsifying drug delivery systems, have been investigated to improve Val bioavailability [42,43,44]. Val-loaded self-assembled mixed micelle systems were also prepared using Pluronic F127 and Tween 80 in order to improve Val oral bioavailability [45]. The influence of Tween 80 on the encapsulation ability of Val by methyl-β-cyclodextrin was also studied [46]. Various terpene-based transdermal drug delivery systems, transdermal gels, monolithic transdermal patches, ethosomes, and nano-ethosomes have also been reported to improve the transdermal delivery of Val [3,5,6,7,47,48,49]. However, self-assembled mixed micelle systems of Val based on the use of Tween 80, Span 80, and sodium dioxycholate (SDC) have not been investigated for their transdermal drug delivery potential. As a result, the objective of this study was to optimize a mixed micelle system for the transdermal delivery of Val in order to improve its therapeutic performance by providing prolonged uniform drug levels while minimizing drug side effects.

## 2. Results

### 2.1. Formulation Factors

Eight formulations were suggested to compare a number of factors including two methods of preparation, micro-phase separation and thin-film hydration, the type of co-surfactants, and the ratio of drug to surfactants. A full description of the exact composition of each formulation is presented in Table 1. All formulations contained Tween 80 as the main surfactant in a combination of either sodium dioxycholate (SDC) or Span 80 as a co-surfactant.

### 2.2. Characterization of the Mixed Micelle Formulations

#### 2.2.1. Particle Size, Polydispersity Index (PDI), Zeta Potential, and Structural Morphology

The average particle sizes for all the prepared formulations were in the nano-range lying between 107.6 nm and 191.7 nm. Table 2 depicts the mean particle sizes of all the prepared formulations. In order to explore the impact of the method of preparation on the micellar mean particular sizes, the results of formulations with the exact same composition, such as the F1 and F6, F3 and F7, and F4 and F8 formulations, were compared. There was no significant difference between the F3/F7 and F4/F8 pairs while a significant difference was observed between F1 and F6. This shows that there is no trend leading to conclusive evidence that the particle sizes were affected by the method of preparation.

Comparing the particle sizes of F5 with F7, F6 with F8, and F2 with F3, it was seen that they differed only in the surfactant:drug ratio, it is notable that the sizes were significantly lower with higher ratios (*p* < 0.05). The effect of using SDC versus Span 80 as a second surfactant with Tween 80 was found to be insignificant with respect to the sizes of the micelles. The F3 formulation (containing 500 mg Span) exhibited a significantly lower particle size (*p* < 0.05) than the F2 formulation (containing 250 mg Span), 112.74 ± 1.73 nm and 137.03 ± 3.42 nm, respectively. This indicates that the ratio of Span 80 is critical for the micelle size. The difference in the type of surfactant, Span 80 (F3 formulation) versus SDC (F4 formulation), did not affect the particle size significantly (*p* > 0.05).

The PDIs for all the prepared formulations are presented in Table 2. The PDI values ranged from 0.24 to 0.39. The PDIs of formulations F1–F4 (0.31–0.39) were significantly higher than formulations F5–F8 (0.24–0.27) (*p* < 0.05). The PDIs for formulations F5–F8 were not significantly different from each other (*p* > 0.05). A cut-off value of 0.3 has been commonly accepted in the literature as the maximum for good size uniformity among a single nanoparticle formulation [50]. Therefore, only the formulations F5, F6, F7, and F8 are considered to have a narrow micelle size distribution. It was also determined that the thin-film hydration method is superior to the micro-phase separation method regarding the uniformity of the micelle sizes.

The zeta-potential values for each formulation shown in Table 2 can be used to estimate the surface charge density of the prepared micelles. The results showed that all formulations had low zeta-potential values, despite differences in their charge nature. The zeta-potential values were significantly different among different formulations (*p* < 0.05). This difference might be due to the use of different surfactant combinations in different formulations. The F6 formulation achieved the highest value (+5.93 mV). Having zeta-potential values above 30 mV has been widely accepted in the literature as a measure of colloidal stability [51]. Other factors, such as the required hydrophilic–lipophilic balance, interfacial tension, and the concentration of the surfactant(s), are more influential to the stability of association colloids such as micelles. Micellar systems are lyophilic (solvent-like) colloids that are stabilized primarily by the formation of a protective solvent sheath rather than by high charge density [52,53].

Figure 1 shows transmission electron microscopy (TEM) images of various micellar samples at various magnification powers. The majority of the particles in image A had particle sizes of around 100 nm. The presence of multiple dark spots within the particles in Figure 1B–D clearly demonstrates drug encapsulation in the core of the particle. The presence of a transparent layer surrounding the micelles (shown by arrows) is attributed to the presence of a solvent-stagnant layer.

The scanning electron microscopy (SEM) image in Figure 2 confirmed the 100 nm average particle size shown in the TEM micrographs which also indicated that the particle shape was spherical. The resistance to degradation by the applied high energy, 100 KV, indicated the prepared micelles’ robust nature, which is unusual for such vesicular particles.

#### 2.2.2. Drug Entrapment Efficiency (EE)

All of the prepared formulations had EE% values greater than 80%, which is an advantage of micellar systems. The formulations with a higher surfactant-to-drug ratio clearly had a higher EE%. Table 2 shows that formulations F3, F4, F7, and F8, with a surfactant-to-drug ratio of 100:1, had a significantly higher EE% (*p* < 0.05), with an average of 94%, compared to formulations with surfactant ratios of 1:45 (F2) and 1:50 (F1, F5, and F6), with an average of 86%. The EE% of an optimized transdermal ethosomal formulation has previously been reported as 80.23% [47], which was much lower than the optimized formulation F7 (96.2%) in this study.

#### 2.2.3. Validation of High-Performance Liquid Chromatography (HPLC) Method

The HPLC method for Val analysis was validated according to the International Council for Harmonization (ICH) guidelines [54]. The representative HPLC chromatograms of pure Val, placebo formulation F7, and final formulation F7 are presented in Figure 3. The HPLC chromatogram of pure Val showed a sharp and intact chromatographic peak at a retention time (R_t_) of 5.66 min (Figure 3A). The chromatographic peak of Val disappeared in the placebo formulation F7 (Figure 3B). The chromatographic peak of Val was retained in the final formulation F7 at the same R_t_, with no additional peaks of excipients (Figure 3B). These results indicated that the chromatographic peak of Val did not interfere with the excipients of formulation F7. The proposed HPLC method was linear in the range of 10–50 µg/mL concentration with a determination coefficient (r^2^) value of 0.9931. The % recoveries of Val were determined to be 99.21–101.31%. The intra-day and inter-day precisions of the method were found to be 0.72–0.87 and 0.78–0.95%, respectively. The limit of detection (LOD) and limit of quantification (LOD) values were recorded as 3.38 and 10.14 µg/mL, respectively. These results suggested that the proposed HPLC method was linear, accurate, precise, and sensitive for the determination of Val.

#### 2.2.4. In Vitro Release

Figure 4A indicates the Val release profile from micellar formulations prepared using the micro-phase separation method (F1 to F4) for the first 24 h. It was found that the rate of release from F1 and F2 is close, with no significant difference over the first 24 h (*p* > 0.05), while F3 and F4 showed a significantly slower rate compared to F1 and F2 (*p* < 0.05). This observation shows that the incorporation of a higher surfactant to Val ratio is the key parameter in delaying the release rate, and the type of co-surfactant had no impact on the Val release from mixed micelles prepared by the micro-phase separation method. Figure 4B displays the Val release profile from micellar formulations prepared using the thin-film hydration method (F5 to F8) for the first 24 h. It was observed that the rate of release from F5 was significantly higher over the first 24 h compared to formulations F6, F7, and F8 (*p* < 0.05).

Figure 5A depicts the Val release profile of micellar formulations prepared using the micro-phase separation method (F1 to F4) for a period from one to 7 days. It is clear that the rate of release of F1 and F2 is very similar, with no significant difference at any point (*p* > 0.05), while F3 and F4 showed a significantly slower rate (*p* < 0.05). This indicates that the incorporation of a higher surfactant-to-Val ratio is the key parameter in delaying the release rate, and the type of co-surfactant had no effect on the Val release from the mixed micelles prepared by the micro-phase separation method. The change in the Val:Span 80 ratio from 1:5 (F2) to 1:20 (F3) caused a significant delay in the release rate. The same pattern was observed with SDC formulations, as F4, which contained a 1:20 ratio, had a significantly slower release rate than F1, which contained a 1:10 ratio.

Figure 5B illustrates the Val-release profile of the formulations prepared by the thin-film hydration method (F5 to F8) over a seven-day period. This result shows that the F5–F8 formulations had controlled release at different rates. After 24 h, the cumulative Val % released was 75% for F5, 58% for F6, 52% for F8, and only 34% for the F7 formulation. According to the findings of this study, a lower drug-to-surfactant ratio slowed the drug release from mixed micelle formulations. F5 had the fastest Val-release among every formulation during the first 12 h, making it suitable for per-oral controlled release because it allows for a gradual release of around 70% of Val within 12 h, enabling a once-daily administration frequency with a maximum fraction of the dose absorbed. During the first 48 h, F6 showed a faster rate of Val release than F8, but the rate of release from both formulations nearly coincided until the end of the study. This is a consequence of the similar surfactant/co-surfactant composition of both formulations and the comparable quantity of drug remaining in both formulations after 48 h. This pattern of resemblance was not observed in formulations containing Span 80 (F5 and F7). This can be explained by the higher hydrophilic nature of SDC-containing formulations when compared to Span 80-containing formulations. The F7 formulation demonstrated an almost constant rate of release, reaching 42% after two days and gradually increasing to 76% after seven days. This slow profile may be advantageous for a variety of delivery systems, including transdermal, long-acting parenteral, and implantable systems. For the first time, our study reports an extremely slow drug release profile for polymeric or small molecule micellar systems. In a recent study, curcumin was combined with cholesterol [55] in an optimal niosome composed of a 7:3 ratio of Span 80:Tween 80. Within 24 h, 75% of the curcumin had been released, according to the researchers. Aboud et al. [45] investigated Val release from mixed micelles composed of Pluronic F127 and Tween 80 in varying ratios for 12 h. They reported that the cumulative % of drugs released from nine formulations ranged from 25 to 60%.

When comparing the Val release profiles of formulations with the same exact composition prepared by different methods, such as F1 and F6, F3 and F7, and F4 and F8, it was notable that there was very good similarity in the release rate for each pair, indicating that the preparation method had no impact and that both methods are suitable for drug incorporation into micelles.

The prepared micelles were found to have remained integral while maintaining a slow drug release profile at 37 °C and continuous shaking for seven days. Similarly, self-assembled Val-loaded polymeric micelles made of poly(D,L-lactide-co-glycolide)-poly(ethylene glycol) block copolymers revealed a nine-day continuous Val release with an average burst release of 20% [56]. Goo et al. [57] found that incorporating Val into a solid self-dispersing micelle composed primarily of Tween 80 and Gelucire 44 increased its release compared to pure Val. The release profile of Val from the mixed micelle systems was similar to those reported previously [56,57].

The kinetics of the Val release from all micelle formulations were studied by fitting to the zero-order, first-order, Higuchi, and Hixson–Crowell equations, as well as determining the Peppas–Korsmeyer (n); the results are shown in Table 3. The r^2^ values clearly indicated their best fit to the Higuchi equation. Korsmeyer et al. [58] and Peppas [59] proposed that n = 0.45 indicates Fickian diffusion, n = 0.46 to 0.88 indicates non-Fickian (anomalous) diffusion, n = 0.89 indicates case-II transport (erosion control and zero-order kinetics), and n = 0.90 indicates case-III transport (erosion control and zero-order kinetics). The calculated n values for all formulations were found to be between 0.385 and 0.442. The Fickian release kinetic model was suggested for the release of Val from the prepared mixed micelle formulations based on the aforementioned criteria. This is consistent with several reports in the literature [60,61]. The absence of any burst release and the slow rate of drug diffusion that decreases with time were observed in all formulations to varying degrees. This demonstrates the Val incorporation in the micelle core and release dependence on the concentration gradient.

#### 2.2.5. In Vitro Skin Permeation Studies

Based on the optimal particle size, optimal PDI, optimal zeta potential, maximum EE, and, most importantly, the sustained and controlled drug release profile, formulation F7 was selected for in vitro skin permeation studies. The in vitro skin permeation profile of Val from the optimized formulation F7 and control is shown in Figure 6. The skin permeation profile of Val from F7, composed of Tween 80/Span 80 micelles, was discovered to be significant compared to control micelles (*p* < 0.01).

In this study, three distinct permeability parameters were predicted from the permeation graphs plotted between the cumulative Val permeated (g/cm^2^) and time (h) (Figure 6). These parameters were the rate of drug permeation via rat skin at steady state (Jss), the permeability coefficient (Kp), and the enhancement factor (E_f_). Table 4 shows the Jss, Kp, and E_f_ values for the F7 micellar formulation and the control. F7 exhibited 16.57 E_f_ as well as Jss and Kp values equal to 68.84 ± 3.96 µg/cm^2^/h and 13.76 ± 0.064 × 10^−3^ cm/h, respectively, which were found to be significant when compared to the control (*p* < 0.01).

Our results were nearly four times higher than the enhancement ratio reported by Ahad et al. [5]. They found that a 15% ethanol carbopol gel formulation produced a 4.53 E_f_ for Val through rat skin. Their reported Val transdermal efflux was 143.51 g/cm^2^/h, which is nearly twice our formulation’s value. In another study, an optimized Val-ethosome formula was tested ex vivo via rat abdominal skin and its antihypertensive effect was tested in vivo in Wistar rats. They discovered a significant increase in transdermal efflux, which was supported by a longer duration of blood pressure-lowering action, when compared to orally administered Val [47]. Similarly, Ahad et al. [48] used the Box–Behnken experimental design to develop an optimized Val ethosome formula containing 35% ethanol. They demonstrated an extremely high efflux through rat skin (801.36 ± 21.45 μg/cm^2^/h). In another study, the use of iso-eucalyptol was shown to enhance Val–skin penetration by a ratio of 7.4 and 3.6 via rat and human cadaver skin, respectively [49].

## 3. Discussion

The incorporation of Val into mixed micelles was intended to overcome its low bioavailability and allow for the transdermal delivery of the drug to provide more uniform drug levels for a prolonged period that consequently enhance its therapeutic outcomes. Micellar-based systems have multiple advantages as a delivery moiety for drugs through the skin. This includes their ability to emulsify a wide range of lipophilic and hydrophilic drugs in hydro-dynamically stable nano-vesicles and their ability to modify the release rate of drugs, in addition to the skin-enhancing property of surfactants [62,63,64].

Mixed micelles have been successfully employed to enhance the transdermal delivery of many drugs including indirubin, arbutin, and diltiazem [65,66,67]. Seo et al. introduced a mixed micellar system composed of a combination of Kolliphor^®^ EL and Tween 80, with polyethylene glycol 400 as a co-surfactant. They showed the effectiveness of their system in the transdermal delivery of an indirubin analog, KY19382 [65]. The dermal delivery of drugs for psoriasis treatment has been approached effectively via their incorporation into mixed micelles [68,69]. Lapteva et al. [68] developed a polymeric micelle nano-system composed of a polylactide-methoxy-poly(ethylene glycol)-dihexyl co-polymer for the dermal delivery of tacrolimus for the treatment of psoriasis. The selective accumulation of micelles into hair follicles was visualized by confocal laser scanning microscopy images. In another study, resveratrol-loaded polymeric micelles were evaluated in vivo in a psoriatic-like plaque mice model, in the form of a gel, and showed significant activity [69].

The selection of Tween 80 as the main surfactant together with either Span 80 or SDC as a co-surfactant was based on their high biosafety, biodegradability, and biocompatibility compared with large molecular weight polymeric surfactants, in addition to their widespread use in food and pharmaceutical products [70,71]; they have been approved by the United States Food and Drug Administration for use in up to 1% of selected foods [72].

The formulations were divided to compare the efficiency of the thin-film hydration and the micro-phase separation methods for mixed micelle preparation. The results indicated that both methods are applicable; however, the thin-film hydration method showed a more uniform particle size distribution and higher drug EE% while the drug release was similar for both methods. This is in compliance with other reports that attribute the widespread use of the thin-film method to its better applicability and higher drug EE, as well as minimal organic solvent residual traces [73,74].

The variation in the drug release profile was dependent on the composition and allowed for variable applications through different routes of administration. Regardless of their vesicular nature, our prepared micelles showed robust integrity, indicating their ability to control the release of Val for a period of one week while withstanding continuous shaking, multiple dilutions, and elevated temperature. Another indication of their integrity and robustness is their resistance to depletion by the high energy (60 KV) and (100 KV) employed for the development of TEM and SEM at high magnifications of 120,000× and 43,000×, respectively. Such an advantage is more common for cross-linked polymeric micelles. Xiong et al. [75] developed a novel mixed micelle composed of two co-polymers containing PCL cores that were shown to have high integrity at very low concentrations and prolonged drug release for more than a week, in addition to their stimulus-triggered targeting ability; stability is a main concern for the success of any micellar drug delivery system [76,77].

In addition to their high integrity and robust properties, our prepared micelles showed a number of interesting attributes, including low molecular size in the range of 100 nm, high EE levels, uniform transdermal release rate over one day, and being entirely composed of safe materials.

## 4. Materials and Methods

### 4.1. Materials

Val was kindly obtained as a gift from Riyadh Pharma Company (Riyadh, Saudi Arabia). Different kinds of surfactants such as SDC, Tween 80, and Span 80 were purchased from Sigma Aldrich (St. Louis, MO, USA). All other reagents and chemicals used were of either HPLC or analytical grade.

### 4.2. Preparation of Val-Loaded Mixed Micelles

For the preparation of Val-loaded mixed micelles, two methods were employed: micro-phase separation and thin-film hydration.

#### 4.2.1. The Micro-Phase Separation Method

The micro-phase separation was performed as previously described by Hu and his colleagues [78]. To summarize, Val and the surfactant mixture were dissolved in dichloromethane to form a true solution. The solution was then added drop by drop to an excess volume of distilled water while being stirred. The organic solvent was completely evaporated after three hours of stirring. To remove any unentrapped drug, the formed micellar dispersion was filtered through a 0.2 m membrane filter.

#### 4.2.2. The Thin-Film Hydration Method

Chen and his colleagues’ [37] procedures were followed. In summary, dichloromethane was used to dissolve Val as well as the surfactant mixture. The solution was dried to form a thin film using an IKA rota-evaporator RV 10 V-C system (IKA-Werke GmbH & Co., Staufen, Germany) at 40 °C + 0.5 and 100 rpm under a reduced 40 mbar vacuum pressure. The organic solvent was completely removed from the thin film by vacuuming it overnight. The dried film was hydrated with 50 mL of de-ionized water pre-heated to 40 °C and stirred for 45 min at 40 °C to form micellar drug dispersion. The dispersion was filtered through a 0.2 m membrane filter to remove any excess drug.

#### 4.2.3. Particle Size, PDI, and Zeta Potential

Samples from each batch were diluted using distilled water to produce a micellar concentration of ~0.1% before processing in a Brookhaven ZetaPALS (Brookhaven Instruments Corporation, Holtsville, NY, USA) to measure the mean particle size and PDI of the size distribution. A 90° angle of detection was used for all measurements. The same instrument used to determine particle sizes was utilized for the zeta potential measurement by applying the laser Doppler velocimetry mode on samples with the same concentration range at 25 °C.

#### 4.2.4. Drug EE

Val-loaded mixed micelle samples were filtered using 0.2 µm membrane filters and then diluted in the methanol. This process was repeated, and then the drug concentration was determined using HPLC by the method previously described by Albekairy and colleagues [79]. The drug EE (%) was determined according to Equation (1).
(1)$$\%EE=\frac{Wt\;of\;initial\;drug-Wt\;of\;free\;drug}{Wt\;of\;initial\;drug}\times 100$$

The HPLC system consisted of an Agilent 1200 series equipped with photodiode array detector of 1260 series (Agilent, Santa Clara, CA, USA). The separation and quantitative determination were conducted utilizing an Eclipsed XBD column (Agilent-PN 993967) C_18_, 150 mm × 3.0 mm i.d., with a particle size of 5 μm. The mobile phase was composed of 46 parts of phosphate buffer (pH 3.6 and 0.01 M), 44 parts of acetonitrile, and 10 parts of methanol. The injection volume was adjusted to 20 μL and a constant flow rate of 1 mL/min at an ambient temperature (25 °C) was maintained during the analysis. Val peaks were detected at 265 nm. The system-integrated software Mass Hunter^®^, version 12.1, as used to automatically calculate the peak areas. Val was eluted at R_t_ = 5.66 min. The proposed method was validated in terms of linearity, accuracy, precision, LOD, and LOQ using ICH guidelines [54].

#### 4.2.5. Particles Morphology

The morphological features of the particles were examined by both TEM and SEM. The TEM measurements were performed using a JEM-1400 electron microscope (JEOL, Tokyo, Japan) operating at an acceleration voltage of 120 kV. A few drops of the F7 formulation were placed on a 400-mesh carbon-coated copper grid. The samples were air-dried at room temperature prior to measurement.

SEM was used to examine the particle surface characteristics of the Val-loaded mixed micelles formulation F7 (JSM-6360 LV, JEOL, Tokyo, Japan). A few drops from formulation F7 were mounted on carbon tape and sputter-coated with a thin gold layer in a high-vacuum evaporator using a gold sputter module (JFC-1100 fine coat ion sputter; JEOL). For scanning and producing photomicrographs of the coated samples, a 10 KV acceleration voltage was used.

#### 4.2.6. In Vitro Release Profile Study

The % of Val released from each mixed micelle formulation was determined by placing a certain amount of the micelle formulation dispersed in 1 mL of phosphate buffer, pH 7, inside a dialysis tube (12 KDa cut-off) that was firmly tied on one end. The dialysis tube was closed then immersed in a vessel containing 50 mL of the same media and placed in a shaking water bath adjusted to 37 ± 1 °C and 80 rpm. A total of 1 mL of each sample was withdrawn at pre-determined time intervals and replaced by fresh, pre-heated medium to maintain the sink condition. The released percentage of Val was determined in each sample using the same HPLC method [79].

#### 4.2.7. Release Kinetic Analysis

The Val-release data were fitted to a zero-order model (Equation (2)), first-order model (Equation (3)), Higuchi diffusion model (Equation (4)), Hixson–Crowell model (Equation (5)), and Peppas– Korsmeyer model (Equation (6)).
Q = k_z_ t(2)
Log Q = Log Q_0_ − k_1_ t/2.303(3)
Q = k_h_ t^1/2^
(4)
(100 − Q)^1/3^ = 100^1/3^ − k_hc_ t (5)
M_t_/M^∞^ = k_p_ t^n^
(6)
where Q_0_ and Q are the cumulative % Val released initially and at time t, respectively; k_z_ and k_1_ are the zero-order and first-order release rate constants, respectively; k_h_ is the Higuchi diffusion release rate constant; K_hc_ is the Hixson–Crowell release rate constant; Mt/M∞ is the fraction of drug released until time (t); and k_p_ is the Peppas– Korsmeyer release rate constant. The exponent (n) in Equation (6) is the slope of the line obtained by plotting log M_t_/M^∞^ (up to 0.6) against log t.

### 4.3. In Vitro Skin Permeation Studies

The in vitro skin permeation profile of Val from different micelles in comparison to the Val suspension (control) was studied using a Franz diffusion cell (FDC). The surface area and volume of FDC were 1.76 cm^2^ and 12 mL, respectively. The rat’s abdominal skin was utilized as a permeation membrane. A Logan transdermal apparatus (SFDC6, Logan Instrument Corporation, Avalon, NJ, USA) was used to assess the skin permeation profile of Val. The skin was excised from the abdominal region of the rat and hair was removed using an electric clipper. The skin was prepared and stored as per the instructions specified in the literature [80,81]. On the day of the experiment, the skin was mounted between the donor and receiver compartments of the FDC, and the procedure was followed as reported in the literature [80,81,82].

Initially, the donor compartment was kept empty and the receiver compartment was filled with 12 mL of freshly prepared phosphate buffer (pH 7). The magnetic bar was included in the FDC. The whole FDC assembly was placed in the Logan transdermal apparatus. The receiver compartment fluid was stirred at 100 rpm and the temperature was fixed to 37 ± 0.5 °C using a thermostat. The whole buffer was replaced at a regular time interval of 30 min in order to stabilize the rat skin. It was found that the fluid in the receiver compartment showed a negligible HPLC response after 6 h and beyond, indicating the complete stabilization of the skin. After stabilization of the skin, 1 mL of Span/Tween micelles and control (each containing 5 mg of Val) were placed into each donor compartment. The donor compartment of each cell was sealed with paraffin film to provide an occlusive environment. An aqueous suspension of the Val was used as the control for the determination of E_f_. Approximately 0.5 mL of aliquots from each formulation was carefully withdrawn and replaced with freshly produced phosphate buffer at regular intervals of 0, 0.5, 1, 2, 3, 4, 6, 8, 12, and 24 h, filtered using 0.45 µm membrane filter, and analyzed for Val content using the same HPLC method [79].

#### Permeation Data Analysis

The cumulative amount of Val permeated via rat skin (µg/cm^2^) was graphed as a function of time (h) for different micelles and the control. The J_ss_ was determined by dividing the slope of the linear portion of the graph by the area of the FDC. The values of K_p_ and E_f_ were determined using Equations (7) and (8), respectively [83,84]:(7)$$K_{p}=\frac{J_{ss}}{C_{0}}$$
(8)$$E_{f}=\frac{J_{ss\;of\;formulation}}{J_{ss}\;of\;control}$$
in which C_0_ is the initial concentration of Val in the donor compartment.

## 5. Conclusions

Val-loaded mixed micelles were successfully prepared using two simple methods that were highly reproducible. The developed micelles had low micellar sizes with a narrow size distribution, high drug EE levels, high integrity, robustness, and prolonged uniform release control with variable rates that can be tuned for multiple routes of administration. Without the use of any skin-modifying enhancers, the best formulation demonstrated extensive transdermal drug delivery through rat skin at a uniform slow rate for 24 h. The prepared system is suitable for a wide range of drugs, particularly those in BCS classes II and IV.

## Figures and Tables

**Figure 1 pharmaceuticals-17-00019-f001:**
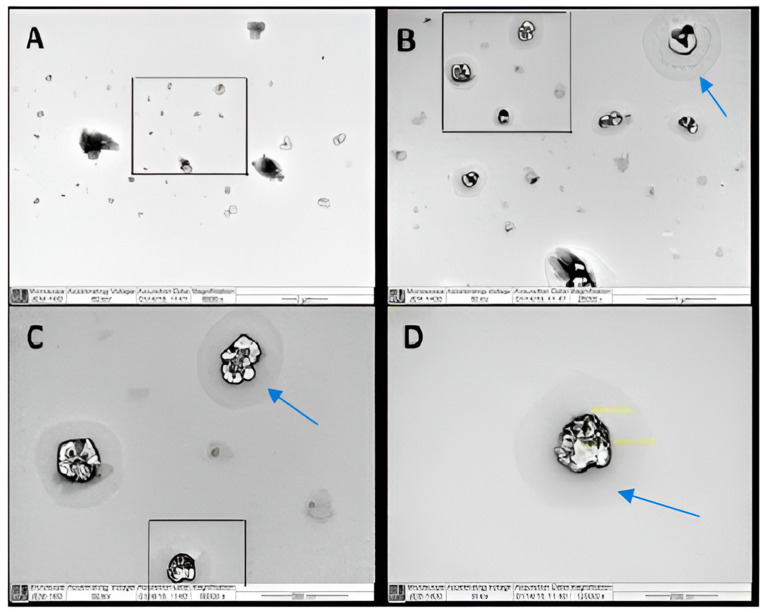
Transmission electron microscopy (TEM) micrograph of valsartan (Val)-loaded mixed micelles (F7); (**A**) a field containing a range of particles at a magnification of 6000×, (**B**) a selected field from image A at a magnification of 25,000×, (**C**) a selected field from image B at a magnification of 60,000×, and (**D**) a focus on one particle from image C at a magnification of 120,000×.

**Figure 2 pharmaceuticals-17-00019-f002:**
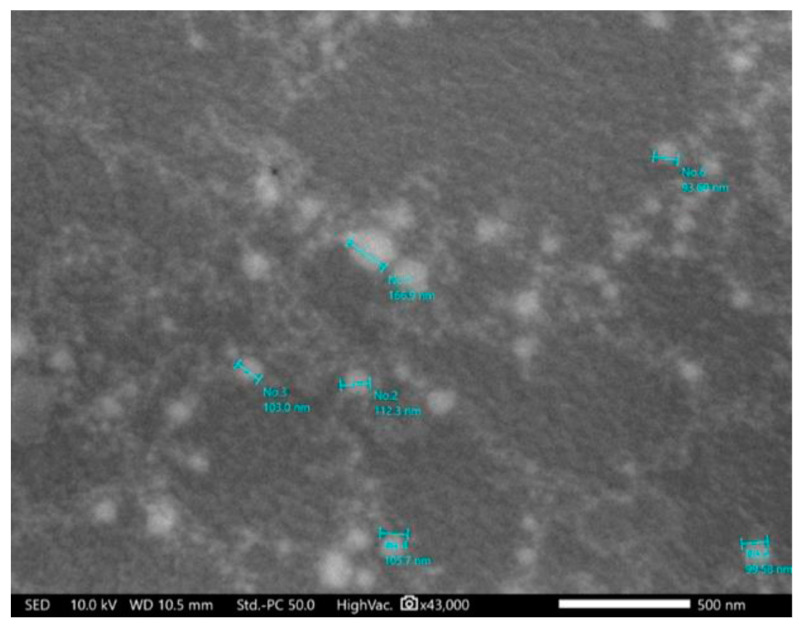
Scanning electron microscopy (SEM) micrograph of Val-loaded mixed micelles (F7).

**Figure 3 pharmaceuticals-17-00019-f003:**
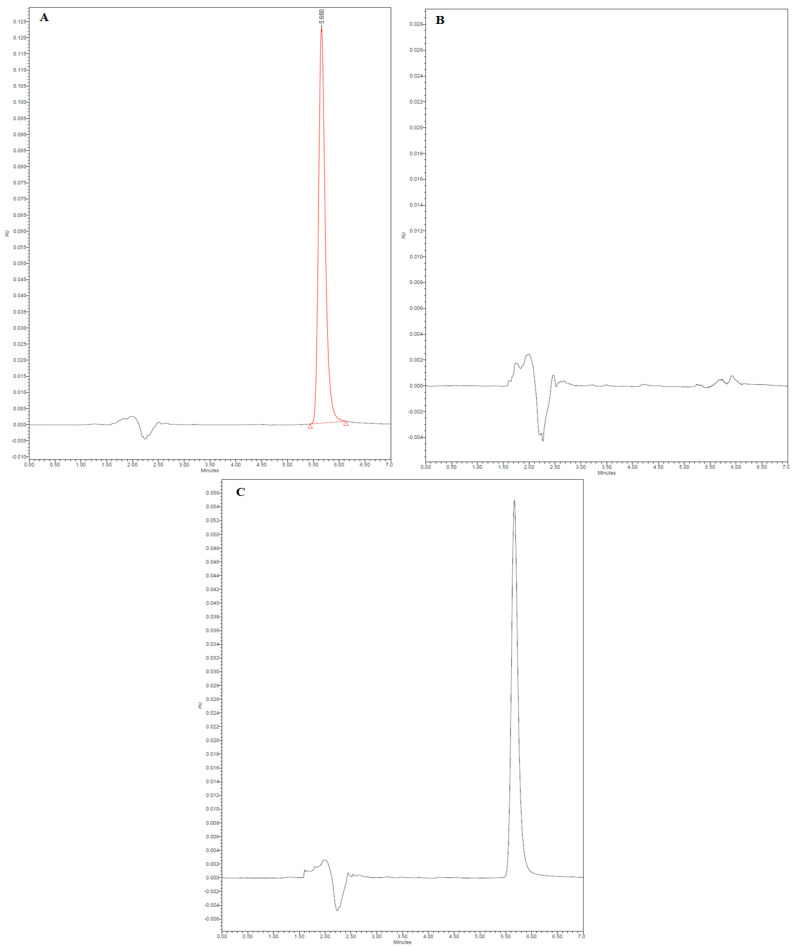
Representative high-performance liquid chromatography (HPLC) chromatograms of (**A**) pure Val (the red line indicates the measured peak area), (**B**) placebo formulation F7, and (**C**) final formulation F7.

**Figure 4 pharmaceuticals-17-00019-f004:**
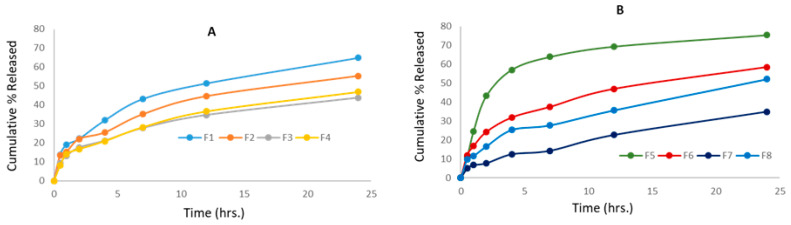
Cumulative percentage (%) release of Val in the first 24 h from all mixed micelle formulations; (**A**) formulations prepared by the micro-phase separation method and (**B**) formulations prepared by the thin-film hydration method (data are presented as the mean ± SD, n = 3).

**Figure 5 pharmaceuticals-17-00019-f005:**
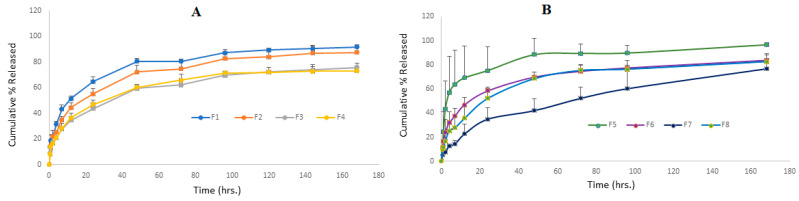
Cumulative percentage (%) release of Val from all mixed micelle formulations against time; (**A**) formulations prepared by the micro-phase separation method and (**B**) formulations prepared by the thin-film hydration method (data are presented as the mean ± SD, n = 3).

**Figure 6 pharmaceuticals-17-00019-f006:**
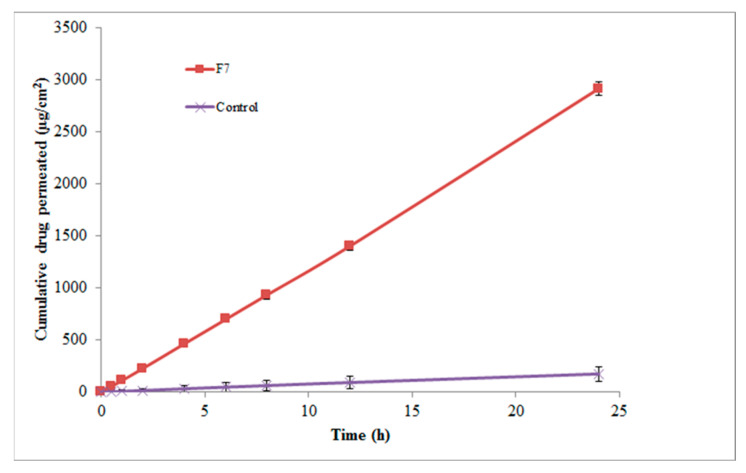
The in vitro skin permeation profile of Val via rat abdominal skin from micelles and an aqueous suspension of Val (control) (data are presented as the mean ± SD, n = 3).

**Table 1 pharmaceuticals-17-00019-t001:** The composition and properties of each of the prepared valsartan (Val) mixed micelle formulations.

Formulation	Val (mg)	Tween 80 (mg)	Span 80 (mg)	SDC (mg)	Method
F1	50	2000	---	500	Micro-phase separation (probe sonication)
F2	50	2000	500	----	
F3	25	2000	500	----	
F4	25	2000	---	500	
F5	50	2000	500	---	Thin-film hydration
F6	50	2000	---	500	
F7	25	2000	500	---	
F8	25	2000	----	500	

**Table 2 pharmaceuticals-17-00019-t002:** The mean particle size, polydispersity index (PDI), zeta potential, and percent entrapment efficiency (EE) measurements for all formulations.

Formulation *	Particle Size (nm)	PDI	Zeta Potential (mV)	EE (%)
F1	107.6 ± 0.6	0.33 ± 0.01	−0.11 ± 0.41	88.1 ± 4.4
F2	137.0 ± 3.4	0.37 ± 0.02	3.74 ± 6.93	82.7 ± 3.3
F3	112.7 ± 1.7	0.31 ± 0.01	−0.67 ± 4.87	95.3 ± 5.7
F4	117.7 ± 7.4	0.39 ± 0.03	0.92 ± 5.90	91.8 ± 4.1
F5	140.4 ± 1.0	0.25 ± 0.00	0.26 ± 7.10	87.3 ± 4.4
F6	191.7 ± 8.5	0.27 ± 0.05	5.93 ± 5.48	86.4 ± 3.1
F7	112.6 ± 0.4	0.24 ± 0.01	−4.93 ± 3.10	96.2 ± 6.7
F8	119.7 ± 0.5	0.24 ± 0.02	3.85 ± 6.34	94.9 ± 2.8

* Mean ± SD, n = 3.

**Table 3 pharmaceuticals-17-00019-t003:** The kinetic parameters of Val release as fitted by various model equations.

Formulation Code	Zero-Order	First-Order	Higuchi Diffusion	Hixson–Crowell	Peppas–Korsmeyer Exponent (n)
F1	0.947	0.979	0.993	0.970	0.409
F2	0.948	0.987	0.995	0.977	0.385
F3	0.935	0.965	0.990	0.956	0.399
F4	0.945	0.977	0.994	0.968	0.423
F5	0.854	0.912	0.935	0.894	0.418
F6	0.937	0.970	0.990	0.961	0.407
F7	0.965	0.986	0.996	0.980	0.524
F8	0.962	0.991	0.997	0.984	0.438

**Table 4 pharmaceuticals-17-00019-t004:** Permeability parameters of the micelles and control.

Formulation	J_SS_ (µg/cm^2^/h) ^a^	K_p_ (cm/h) ^a^ × 10^−3^	E_f_
Control ^b^	4.15 ± 0.37	0.83 ± 0.016	*-*
F7 (Span/Tween micelles)	68.84 ± 3.96	13.76 ± 0.064	16.57

^a^ Mean ± SD, n = 3, ^b^ aqueous suspension of Val was used as control.

## Data Availability

Data is contained within the article.

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
