# Peer review of "Tween 80-Based Self-Assembled Mixed Micelles Boost Valsartan Transdermal Delivery"

_pharmaceuticals, 2023, doi:10.3390/ph17010019_

Round 1

Reviewer 1 Report

Comments and Suggestions for Authors

The manuscript titled “Tween 80-based self-assembled mixed micelles boosted valsartan transdermal delivery” by Yassin et al. describes the in vitro drug release properties of valsartan-loaded Tween 80 micelles for transdermal drug delivery.The work is generally well described and the materials well characterized. However, I have strong concerns related to the novely of the proposed approach, since drug delivery carriers for the transdermal delivery of valsartan have been widely explored in the literature starting from 2013. For example, see:

https://link.springer.com/article/10.1208/s12249-012-9865-5

https://www.tandfonline.com/doi/abs/10.3109/08982104.2012.753457

https://www.tandfonline.com/doi/full/10.2147/IJN.S136599

https://link.springer.com/article/10.1208/s12249-015-0388-8

Not only, but the combination of valsartan with Tween 80 for the preparation of self-assembled micelles has been reported in 2020: https://www.tandfonline.com/doi/abs/10.1080/1061186X.2019.1650053 and https://www.sciencedirect.com/science/article/pii/S0144861713012617

A thorough and detailed comparison of the proposed approach with previously reported results must be included in the work, in order to identify (if present) any significant advantages of the proposed approach compared to the many alternatives presented in the literature.

For these reasons, I believe the proposed approach does not represent any significant advancement in the field, and I recommend the editor of Pharmaceuticals for the rejection of the manuscript in its current form due to its complete lack of novelty.

Author Response

I would like to thank you for the effort spent in revising the manuscript. All your query and comments are highly appreciated and I believe that they have helped us to improve the quality of the manuscript. I have responded to all of them and made modification to the manuscript accordingly.

Reviewer 2 Report

Comments and Suggestions for Authors

Dear Author,

I would like to inform you that the manuscript entitled “Tween 80-based self-assembled mixed micelles boosted valsartan transdermal delivery” has been intensively reviewed and evaluated. The manuscript was found well designed and structured, however there were several points that need to be revised. After the revision process, it could be a good candidate for scientific literature.

Hereby I would like to present my comments:

Comment_1: Generally, the author has preferred to use abbreviations, please carefully keep the harmony of your abbreviations. (abbreviations had been used in the first place then unabbreviated forms used, please check whole text and carefully revise all of them) (some examples: (“valsartan” abbreviated as val (L31, L43) and unabbreviated forms (L141, 212, 247, 290, 291, 295…etc.), “entrapment efficiency” abbreviated as EE (L36) and unabbreviated forms (L37, 152,354…etc.), “polydispersity index” abbreviated as PDI (L37) and unabbreviated forms (L144, 145…etc.).

Comment_2: The Latin words should be written italic (in vitro, in vivo, Stratum corneum…etc.), please check whole manuscript and revise them.

Comment_3: check the typo “United State” should be “United States”.

Comment_4: Please check the introduction part L101-108. Provide a connected and consistent paragraph, the first two sentences remain irrelevant.

Comment_5: In Table 2 the Asterix in F1 is only specific for the related formulation or includes all the formulations? If includes all the formulations, Table presentation could be revised.

Comment_6: In several times, the authors mentioned that micelles demonstrated high stability, good physical stability…etc. even though, there were no long term physical or chemical stability tests conducted. How could they prove the high stability of formulations (only 7-day release is not considered as high stability (as a general term)). Please provide short or long term stability results (60 days to 180 days) or make a reasonable explanation or revise the expressions in a rational perspective.

Comment_7: Please demonstrate the chemical interactions of formulation ingredients (provide FTIR (which is a universal instrument in research laboratories) data: API, final formulation, and physical mixture…etc.). Then, please interpret the data obtained.

Comment_8: Please provide a high-resolution images of transmission electron microscopy (TEM) and if it is not possible, please provide the magnification levels and scale bars.

Comment_9: Please provide an extra figure for the observation of first 24h in Figure 4.

Comment_10: In Table 3, why the first order kinetics were excluded? Please provide first order release kinetics. Also check the name of “Higuchi”.

Comment_11: As a major point (critical request, please carefully provide the requested data), the analysis method was not validated. The authors might consider that it had already been developed method, but they must demonstrate that the quantification method is specific for this formulation. As an integral part of analysis, the chromatogram of valsartan has not interfered with the other components of formulation. Please provide the chromatograms of valsartan (as a pure drug substance), placebo formulation and the final formulation. Moreover, provide LOD and LOQ data for the determination of lower limits of your analysis.

Best regards.

Author Response

I am very grateful for the effort spent in revising the manuscript. All the comments and suggestions were highly appreciated and I believe that they have helped us to improve the quality of the manuscript. I have responded to all of them and amended the manuscript in many places accordingly.

Reviewer 3 Report

Comments and Suggestions for Authors

The authors have successfully manufactured Tween 80-based micelles for transdermal delivery of valsartan. Generally, this manuscript is very interesting. A minor English editing is required for this article. Here are some points for improving the manuscript:

1. Abstract: This has been well written, but with few points should be considered.

- The authors mentioned the side effects. Are these side effects related to the next sentences? Is it important to mention these aspect in the first line of the abstract?

- "The goal here is to incorporate ..." If this statement is the research aim, this has to be in Past tense form. Please replace term "goal" with aim or purpose to sound more scientific.

2. Introduction:

- Page 2 Line 51: "Low oral bioavailability ..." This statement can be associated with BCS Class of Val.

- Page 2 Line 53: "adverse effect" Again, are these side effects related to the low bioavailability and short half-life of the Val? Please provide more explanation on this.

- Page 2 Line 56: The authors did not mention clearly the weakness of oral Val and the urgency to modify the delivery system to transdermal one.

- Page 3 Line 101: The author should add some data about previous studies on the utilization of micelles for delivering antihypertensive drug transdermally

- Page 3 Line 109: This goal/aim should appear in the abstract.

3. Results:

- Page 3 Line 124: Figure 1 should appear before Table 2. Please rearrange the order of the Figure and Table.

- Page 3 Line 133: The effect of SDC versus Span on the particle size. This statement should be explained with the suitable theory 

- Table 1: In this Table, why did F2 only contain 250 mg Span 80? There is no comparison to this formulation. The authors should justify the reason of choosing this different amount of Span.

- Page 4 Line 144: Is there any significant difference on the PDI? Please write the discussion about this and explain the relationship between PDI and method used.

- Table 2: There should be discussion about the significant difference on zeta potential values of the formulations prepared.

- Figure 1: The authors should choose between Tables or bar graphs for explaining the results. If the data presented are the same, it is better for authors delete either the table of figures.

- Page 5 Line 175: Explaining the solvent stagnant layer: This has to be pointed out in the Figure. Please give more annotations on the SEM Figure.

- Figure 3: This Figure is quite blurry and unclear. Please provide a better image for this one.

- Page 7 Line 194-196: Please discuss about this and mention some other works which are similar to this research.

- Figure 4 & 5: Is it possible to merge Figure 4 and Figure 5? It will be easier for the readers to compare them in one graph.

- Figures: Please provide the samples number and how the data are displayed, for example: Mean plus minus SD, n = X.

- Page 9 Line 250: "very good physical stability" Did author perform stability study on the fabricated micelles?

- Page 9 Line 255:  "Tween 80 and Gelucire 44 increased its oral bioavailability" This statement is not appropriate here because the authors did not conduct an in vivo release study.

- Figure 6: Please explain the reason in choosing only F7 which was continued to the in vitro permeation study?

4. Discussion: Please add the discussion with the comments provided above and compare the results obtained in this research with the work of others when relevant. If possible, it will be better for authors to merge results and discussion sections together for better readibility.

5. Materials & methods: Good and clear.

6. Conclusion: Please add some statements regarding the future aspects of the research. Regarding the explanation of BCS class II and IV, this should be mentioned earlier in the background.

Comments on the Quality of English Language

Minor editing of English language required

Author Response

I appreciate the time and effort you put into revising the manuscript. All of the comments and suggestions were greatly appreciated, and I believe they helped us improve the manuscript's quality. I responded to all of them and made numerous changes to the manuscript as a result.

Round 2

Reviewer 2 Report

Comments and Suggestions for Authors

Dear Author,

I have reviewed the revised version of the manuscript you submitted, and significant changes have been performed. These modifications have been evaluated as successful in making the manuscript a candidate for publication. I congratulate you and your team on the success of this work. I am pleased to express a positive opinion of the publication of your manuscript.

Best regards,

Reviewer 3 Report

Comments and Suggestions for Authors

Thank you for addressing the comments very well. This manuscript has been greatly improved, and is now acceptable for publication.